# Biosensor libraries harness large classes of binding domains for construction of allosteric transcriptional regulators

Javier F. Juárez[1,3], Begoña Lecube-Azpeitia[1], Stuart L. Brown[1], Christopher D. Johnston [2] & George M. Church [1]

The ability of bacteria to sense specific molecules within their environment and trigger metabolic responses in accordance is an invaluable biotechnological resource. While many transcription factors (TFs) mediating such processes have been studied, only a handful have been leveraged for molecular biology applications. To expand the repertoire of biotechnologically relevant sensors we present a strategy for the construction and testing of chimeric TF libraries, based on the fusion of highly soluble periplasmic binding proteins (PBPs) with DNA-binding domains (DBDs). We validate this concept by constructing and functionally testing two unique sense-and-respond regulators for benzoate, an environmentally and industrially relevant metabolite. This work will enable the development of tailored biosensors for novel synthetic regulatory circuits.

[1] Department of Genetics, Harvard Medical School, 77 Av. Louis Pasteur, Boston, MA 02115, USA. [2] The Forsyth Institute, 245 First St., Cambridge, MA 02142, USA. [3] Present address: The Forsyth Institute, Cambridge, MA 02142, USA. These authors contributed equally: Begoña Lecube-Azpeitia, Stuart L. Brown. Correspondence and requests for materials should be addressed to G.M.C. (email: gchurch@genetics.med.harvard.edu)

Synthetic biology is a rapidly evolving discipline[1], propelled by advances in the synthesis and assembly of ever longer and more complex DNA sequences[2]. Since the turn of the century, progress in DNA synthesis has been accompanied by continued discovery, characterization, and adaptation of novel systems for regulation of gene expression, such as riboswitches[3], TALENs[4,5], and RNA-guided nucleases (CRISPRi-dCas9)[5,6]. Nevertheless, monogenic transcription factors (TFs), which regulate gene expression upon binding of a soluble small molecule known as an inducer, remain the workhorses of the gene regulation world. For decades, robust bacterial transcriptional repressors such as LacI[7] and TetR[8] have been the preferred TF choice, pairing an inducer to common reporters (e.g., antibiotic resistance, green fluorescent protein (GFP) and LacZ) by controlling the promoters that drive their expression. When compared to two component signal transduction systems, the arrangement of sensor and effector in one molecule is simpler and more effective[9], making monogenic TFs ideal[10] for whole-cell biosensor applications[11]. While two component systems can detect external molecules that are unable to traverse the cell envelope, their use as biosensors is limited by the risk of crosstalk between systems[12]. Yet, despite their advantages only a small number of monogenic TFs are available[13]. The rational design of synthetic monogenic TFs that respond to small molecules of interest has been a long-term aspiration of synthetic biologists and would be tremendously useful for biotechnological applications[10,14].

Here, we focus exclusively on the development of monogenic intracellular sensors to avoid the undesired traits of two component systems, such as their expression as membrane proteins and the risk of activation by unspecific phosphorylation. Our emphasis is on the generation of new TFs capable of detecting small molecules. It is noteworthy, however, that key aspects for the fine tuning of their expression, as well as the refinement of their dose–response curves and ligand affinity, are not tackled in this study. Nevertheless, the products of our assembly and enrichment process are the ideal substrate for systematic expression improvement strategies[15,16]. In this work, we present a high-throughput pipeline for in vitro construction and in vivo testing of tailor-made transcriptional regulators by massively multiplexed fusion of protein domains and linkers[17]. This approach is validated by the generation of two new benzoate-binding TFs.

Despite 3 decades of research demonstrating that TFs originate from the fusion of individual gene modules[18], a general method to create functional fusions of two gene domains has remained an elusive holy grail, due to easily broken allosteric interactions between proteins[19]. LacI/GalR regulators can remain active when their ligand-binding domains (LBDs) are swapped with members of their protein family[20]. Their DNA-binding domains (DBDs) recognize the same operators, but the new TFs instead respond to the fused LBDs inducer molecule. As DBDs of the LacI/GalR family originated from periplasmic binding proteins (PBPs) that recognize sugars[21], there have been attempts to create a novel biosensor by substitution of LacI-LBD with a PBP. The main example is SLCP$_{GL}$: a glucose-responsive TF built by the fusion of E. coli galactose/glucose binding protein (GGBP) to DBD-LacI[22]. The chimeric TF Q1 is another example of the generation of a new TF by the fusion of a DBD (from BzdR) to a protein phylogenetically related to its LBD (shikimate kinase)[23]. The paucity of novel TFs created by fusion of DBDs to proteins that are not integral parts of regulators emphasizes the challenge of de novo generation of new biosensors.

For systematic construction of fusion TFs we generated libraries under the following two degrees of freedom: (a) 15 DBDs sourced from bacterial transcriptional repressors with a common architecture and known operator sequences; and (b) 15 LBDs from PBPs associated with ATP-binding cassette transporters. Figure 1 summarizes our approach for generation of novel TFs described in this work. This pipeline required the development of new gene assembly methods as well as the construction of a collection of tailor-made reporters. The two novel TFs presented here are based on the fusion of DBDs to LBDs that were not part of preexisting regulators, but rather independent proteins with the ability to bind soluble small molecules. DBD and LBD were connected through a series of linkers (LNKs).

All TFs resulting from this combination of domains were functionally tested with benzoate, the inducer molecule specifically recognized by a group of LBDs included within the library. Benzoate was selected for this proof of concept work due to its environmental relevance, as it is associated with the degradation of lignin: the second most abundant polymer on earth after cellulose[24,25]. The reductive fractionation of lignin frees abundant benzoate-related aromatic compounds that can be used as building blocks for applications such as biofuel production[26]. Additionally, Escherichia coli is unable to degrade benzoate and it is relatively nontoxic[27], allowing for its direct addition to culture media[28]. BenM is a transcriptional activator that can simultaneously bind benzoate and cis,cis-muconate, but its applicability as biosensor is not ideal due to its complex specificity profile[29,30]. In this context, the creation of a chimeric TF capable of specifically sensing benzoate is a viable alternative that can demonstrate the potential for systematic construction of custom-made TFs.

## Results

**Design of transcriptional regulator library**. We used a novel chimeragenesis method for the construction of custom TFs for benzoate-binding. This method is based on the generation of a library of repressors each with a common architecture: Nt-DBD-(LNK)-LBD-Ct. All DBDs were selected from repressors that exert control over promoters recognized by the σ$^{70}$ subunit of RNA polymerase and thus carry $-10/-35$ operator boxes[31]. The transcriptional repressors that inspired these chimeras all share a common modular organization: an N-terminal (Nt) domain responsible for DBD and a C-terminal (Ct) domain that recognizes a specific compound (LBD), which are either directly connected or associated by a linker (LNK) region that tended to be less structured than the domains they connected[32]. The native LBD of the TF, which normally confers the specificity for the inducer molecule is replaced by a protein with the potential to recognize the desired molecule of interest (i.e., benzoate/4-hydroxybenzoate). PBPs were selected to serve as LBDs owing to multiple advantages: (a) a high solubility, stability, and affinity for their substrates;[33] (b) a detectable conformational change when bound to their ligands[34], which is key to altering the oligomerization state of the TFs and thus their ability to bind DNA;[35] and (c) a phylogenetic relationship with the LBD of the LacI/GalR family[36].

Each of the resulting chimeric genes carried a DBD, a LBD, and in some cases a LNK domain. Different criteria were set for these three sets of domains with a common goal of maximizing diversity. For DBDs, this goal was achieved by introducing distinct three-dimensional (3D) configurations, while for the LNK domain we utilized different lengths and structural flexibilities. In the case of the LBDs, selection was based on a concise list of proteins with demonstrable binding affinity for benzoate/4-hydroxybenzoate. This list was expanded to include analogous and homologous proteins according to amino acid sequence. Table 1 contains the transcriptional repressors whose DBD were used for the construction of the chimeric TFs, whereas Table 2 lists the proteins chosen as LBD. The LNK domains

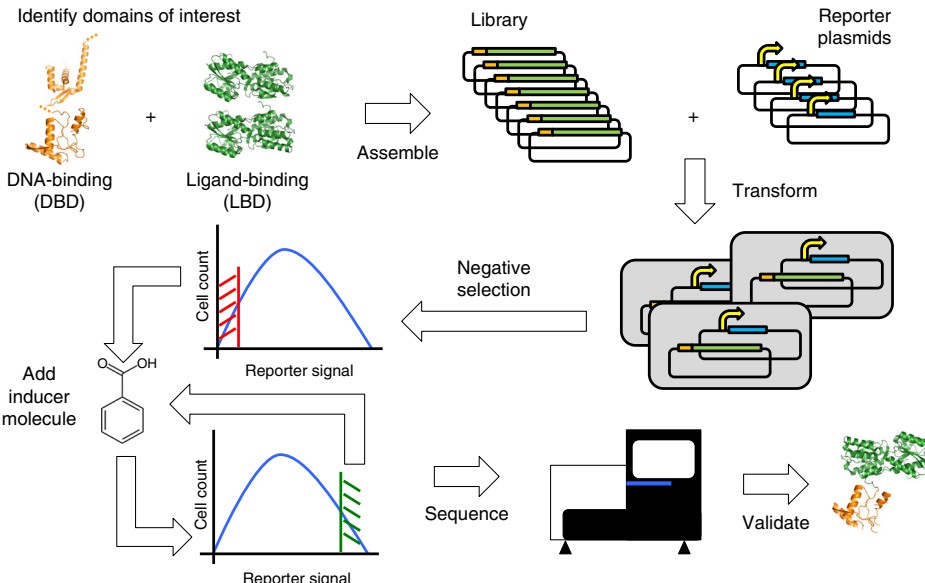

**Fig. 1** Pipeline for the generation of novel TFs through gene fusion. Starting top-left, the process starts with the selection of the DNA-binding domains (DBDs), regions from preexisting TFs that interact with DNA on specific operator sequences, and the ligand-binding domains (LBDs) that provide the chimera with inducer specificity. Libraries of TFs are assembled combining DBDs and LBDs. In parallel, a set of reporter plasmids was developed. Each reporter plasmid contained a modified *Ptac* promoter in which LacI operator boxes had been replaced by operator sequences for one of the DBD included in the collection of chimeras. Both the library of chimeras and the library of reporter plasmids were transformed into *E. coli* host cells. A first negative selection is performed using FACS to enrich the libraries in properly paired DBD/reporter plasmids: bacteria containing chimeras whose DBD were unable to recognize the operator boxes included in the reporter plasmid could not repress the transcription of a reporter gene encoding GFP. Non-GFP-fluorescent bacteria were recovered and subsequently exposed to the inducer molecule (benzoate). In this case the chimeras that were able to derepress the reporter promoter allowed the transcription of GFP, facilitating the positive discrimination of inducible chimeras. This positive selection was iterated and the relative abundance of every chimera compared before and after the selection. Both the starting and final libraries were sequenced, assessing the relative abundance of every chimera. Candidate TFs enriched in the final library were individually assayed in vivo. Protein structures in orange and green: PDB 1MJL, 3F8C, and 2HPH

| Table 1 DNA-binding domains included in the chimeric TFs | | |
| --- | --- | --- |
| **DNA-binding domains (DBD)** | | |
| **NAME** | **FAMILY** | **ACCESSION** |
| ArgR | ArgR | AAK45964 |
| FL11 | AsnC/Lrp | BAA30629 |
| DeoR | DeoR | AAC73927 |
| LacI | LacI/GalR | AAC73448 |
| TreR | LacI/GalR | Q2M666 |
| FadR | GntR | AAC74271 |
| TtgV | IclR | AAK69562 |
| CbnR | LysR | BAA74529 |
| CueR | MerR | AAC73589 |
| MetJ | MetJ | AAA24163 |
| ModE | ModE | AAB06892 |
| LmrR | PadR | CAL96929 |
| TtgR | TetR | AFO46103 |
| QacR | TetR | ADK23698 |
| Xre | Xre | AAA22894 |

incorporated as DBD/LBD bridges can be found in Supplementary Table 2. Beyond LNK domains, another source of diversity was the inclusion of variants for each DBD in the library. To compensate for the lack information on the functional boundaries of the DBD domains, we introduced multiple variants of each DBD. Each variant differed by the length of the DBD domain taken from the native DBD/LBD amino acid sequence. The shorter version of every DBD was designated as "CORE", while the extended variants became "ENDS". On average there were eight "ENDS" per DBD. Concurrently, two versions of each LBD

were included: one with and one without the native signal peptide (SP and nSP, respectively). A complete description of the selection process and restrictions applied to the final list of modules for the chimeragenesis method can be found in Supplementary Note 1 accompanied by expanded tables including additional information on the 3D structure, size, and limits of the selected DBD and LNK domains (Supplementary Tables 1 and 2).

Finally, to improve allosteric interactions, an additional domain of LacI was included in the construction of the library. The functionality of LacI not only depends on its ability to associate into oligomeric states, but also on the duration that it can maintain these estates[37]. LacI contains an oligomerization domain (OD) within its Ct that promotes the association of two LacI dimers, creating de facto a tetramer and allowing it to remain in a prolonged dimerized state, which increases its repressing efficiency[37]. We included LacI OD in the design of the TFs presented here so that two versions of each chimera were created: one without and one with OD (translationally fused to the Ct region). Supplementary Note 1 contains further information on the inclusion of LacI OD within synthetic TFs.

To identify individual chimeras generated in this work, we adopted a nomenclature to provide information on the succession of their elements, from Nt to Ct: DBD-(LNK)-LBD-(OD). Brackets indicate components that may or may not be present in individual chimera. As an example, the aforementioned chimeric regulator $SLCP_{GL}$[22] is designated as LacI-GGBP-OD.

**Novel multiplexed assembly to create a library of regulators.** To effectively construct thousands of chimeric TFs by gene fusion, we first created a systematic pipeline to design and build them. A

**Table 2 Ligand-binding domains included in the chimeric TFs**

**Ligand-binding domains (LBD)**

| NAME | ACCESSION | LIGAND | COMMENTS | SIGNAL PEPTIDE (AA) |
|------|-----------|--------|----------|---------------------|
| GGBP | AAC75211 | Glucose(*) | Internal Positive Control | 1–23 |
| RPA0668 | CAE26112 | Benzoate | Empirically proven:[70] | 1–31 |
| RPA4029 | CAE29470 | 4-OH-benzoate | Empirically proven:[70] | 1–27 |
| RPA0985 | CAE26428 | 4-OH-benzoate | Empirically proven:[70] | 1–22 |
| ADP71087 | ADP71087 | Benzoate(P) | PID 70% to RPA0668 | 1–31 |
| KAI94709 | KAI94709 | Benzoate(P) | PID 70% to RPA0668 | 1–31 |
| ABE44898 | ABE44898 | Benzoate(P) | PID 53% to RPA0668 | 1–30(E) |
| ABE38823 | ABE38823 | 4-OH-benzoate(P) | PID 89% to RPA4029 | 1–27 |
| EYC50849 | EYC50849 | 4-OH-benzoate(P) | PID 68% to RPA4029 | 1–25 |
| ABD68043 | ABD68043 | 4-OH-benzoate(P) | PID 65% to RPA4029 | 1–24 |
| AHF85493 | AHF85493 | 4-OH-benzoate(P) | PID 65% to RPA0985 | 1–20 |
| CAK09396 | CAK09396 | 4-OH-benzoate(P) | PID 66% to RPA0985 | 1–20 |
| AEK56128 | AEK56128 | 4-OH-benzoate(P) | PID 63% to RPA0985 | 1–29 |
| BzdB1 | AKU11518 | Benzoate(P) | Associated to benzoate catabolism | 1–30 |
| BAE51678 | BAE51678 | Benzoate(P) | Associated to benzoate catabolism | 1–41 |

(*) Built-in control, (P) Putatively, and (E) Estimated; signal peptide identified by similarity to other proteins, not easy prediction based on software described in Section Methods; PID percentage of identity (NCBI Protein BLAST)

general outline of the assembly strategy employed for the construction of TF libraries can be found in Fig. 2. Briefly, we started by designing a set of 4275 "core chimeras" (15 DBDs × 19 LNKs × 15 SBPs). These chimeras are a microcosm of the whole library, representing all the fundamentally different gene domains included in the TFs. The actual maximum number of different fusion genes that could be generated in the library is 135,660. These are the chimeras that could be generated by assembling combinations of 119 DBD-ENDS, 19 LNK/no LNK, 30 Ct-LBD, and 2 OD/no OD. To simplify the synthesis of these fusion genes they were first assembled without an OD domain and subsequently cloned into two expression vectors, pCKTRBS and pCKTRBS-OD (Methods, Supplementary Data 1).

The TFs cloned into pCKTRBS-OD incorporated a 3′-translational fusion to an OD sequence included within the plasmid chassis, whereas the TFs cloned into pCKTRBS did not. Each of these two libraries could potentially contain a maximum of 67,830 chimeras (assuming cloning efficiency of 100% and their expression was nontoxic to the cells harboring them[38]). Supplementary Fig. 1 represents the true size distribution of the designed chimeric TF genes compared to their abundance in the library.

To connect Nt-DBD and Ct-LBD domains with a diverse set of LNK sequences, we developed a novel cloning strategy based on the ligase cycling/chain (LCR) reaction assembly method[39]. We refer to this new method as "enhanced LCR" or eLCR (Methods). Conventional LCR provides scarless fusion of dsDNA fragments. Our eLCR method allows the targeted insertion of oligonucleotides in the junction between those fragments, enabling the generation of many different connections between DBD and LBD domains. Each DBD in the library was composed of a fixed sequence identified as DBD-CORE that needed to be connected to the LBD. These two domains were bridged by one or both of two elements: (i) specific extensions of every DBD-CORE called ENDS and (ii) LNK sequences. eLCR enabled the use of oligonucleotides to attach all the possible ENDS to their correspondent DBD-CORE plus a LNK option per chimera. Fig. 2 shows a schematic representation of the different classes of chimeras present in the library attending to the number of oligonucleotides necessary to assemble them. The rationale for this design is detailed in Methods and Supplementary Methods.

The fusion genes assembled by eLCR were PCR amplified with oligonucleotides that incorporated adapter sequences

homologous to the expression vectors pCKTRBS or pCKTRBS-OD (Supplementary Data 1). The adapter to pCKTRBS-OD created a translational fusion to the LacI OD integrated in the vector. All oligonucleotides utilized are listed in Supplementary Data 2. The bacterial libraries expressing the chimera collections from pCKTRBS/pCKTRBS-OD were named *E. coli* (Ch-END) and *E. coli* (Ch-OD) (Supplementary Data 1). To estimate the composition of the library we sequenced a representative sample of the chimeric TFs within in *E. coli* (Ch-END) and *E. coli* (Ch-OD) libraries (Methods). Supplementary Table 3 summarizes the assignment of fusion genes contained in both libraries to the different classes of chimeras described in Fig. 2 and Supplementary Note 1. Chimeric TF genes containing every DBD and LNK were found in different abundance. However, four LBDs did not integrate into any chimeras (ADP71087_nSP, AHF85493_nSP, CAK09396_nSP, and KAI94709_nSP). We were unable to discern whether these were unfavored in the cloning process or if the fact that these four LBD lacked their SP increased the toxicity of the chimeric TFs after each had integrated.

**Design and construction of synthetic reporters**. We designed a reporter system based on the controlled expression of GFP by synthetic promoters to screen the activity of chimera libraries. To assay the activity of a given chimera, a reporter plasmid specific for the DBD of said chimera was required. Having 15 DBD modules in the assembly mix, we constructed 14 modified *Ptac* promoters (and one additional for *Ptac* itself), each engineered to contain operator boxes for a specific DBD. The sequence of these 15 promoters is shown in Supplementary Data 3. The high-copy number family of reporter vectors pHC_DYO*DBD*-R (Supplementary Data 1) was constructed by integrating the 15 promoters into a pUC chassis, so that each controlled the expression of GFP (Methods). In the presence of chimeric TFs that retained their DNA-binding ability, expression of the GFP reporter gene should decrease. In contrast, upon addition of the benzoate inducer molecule to the culture, repression of the GFP reporter should be relieved if the chimeras are functional.

The 15 synthetic promoters that were integrated into pHC_DYO*DBD*-R plasmids all required an approximate basal expression level, to minimize the intrinsic bias of the pipeline toward the enrichment of TFs whose DBD interacted with the strongest promoters. Several versions of the modified promoters

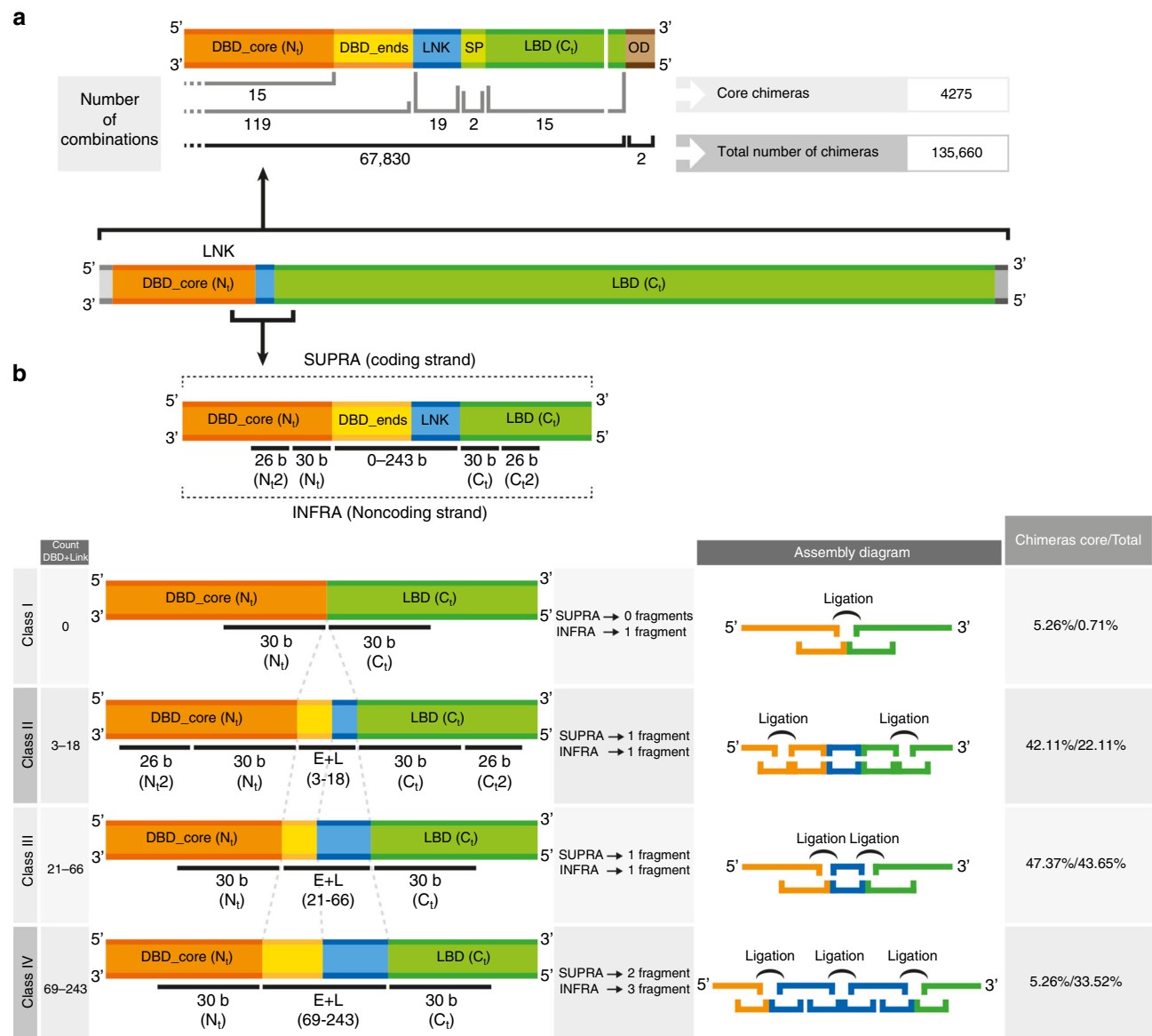

**Fig. 2** Domain structure of chimeric TFs and complexity-dependent classification for eLCR. Representation of an average chimera maintaining the proportion between the lengths of the different regions that integrate it. DBD-CORE are represented in orange, ENDS are represented in dark yellow, LNK in blue, signal peptides (SP) in light green, LBD in dark green, and OD in brown. **a** Not-to-scale representation of the joint between DBDs and LBDs together with OD. The number of domains of each kind incorporated into the assembly scheme as well as the maximum number of chimeras resulting from the eLCR are also represented. **b** Zoomed in simplified version of the joint between DBDs and LBDs. ENDS and LNKs occupy the center of the representation, as they were the regions incorporated in the eLCR through a collection of oligonucleotides. DBDs and LBDs (with and without SP) were introduced in the assembly as dsDNA fragments. The numbers below the different domains represent the number of DNA bases included into the oligonucleotides in the DNA microarray synthesis. The panel below represents the number of bases corresponding to each one of the domain types that needed to be introduced into the eLCR oligonucleotides depending on the assembly Class the chimera was assigned to. The Assembly Diagram represents, in the same color code that above, the final Supra and Infra oligonucleotides and the ligation events that needed to take place to reconstitute the final chimera. Finally, the percentage of "CORE" and total chimeras assigned to each one of the Classes, in the library design, is included on the right side of Panel B

were created and inserted into pHC_DYO*DBD*-R. The basal expression level of each compared to *Ptac* was empirically assessed in vivo (Methods). Synthetic promoters with activity outside a threshold value of one order of magnitude were discarded. Supplementary Figure 2 shows the relative basal expression of the final set of synthetic promoters compared to *Ptac*. To supplement our individual assessment of activity, a mixture of strains carrying all reporter promoters was assayed using FACS. This demonstrated that when the strain carrying the wild-type promoter *Ptac* was removed from the mixture,

remaining strains were not preferentially enriched based on their basal GFP expression (Methods). Subsequently, *E. coli* (Ch-END) and *E. coli* (Ch-OD) were transformed with the collection of reporter plasmids as described in Methods. The resulting *E. coli* libraries, in which each bacterium expressed a chimera and carried a reporter promoter, were identified as AYC Lib-Ch-END and AYC Lib-Ch-OD, respectively (Supplementary Data 1). TFs with leaky promoters, where basal expression led to toxicity for the *E. coli* host, autonomously eliminated themselves from the libraries.

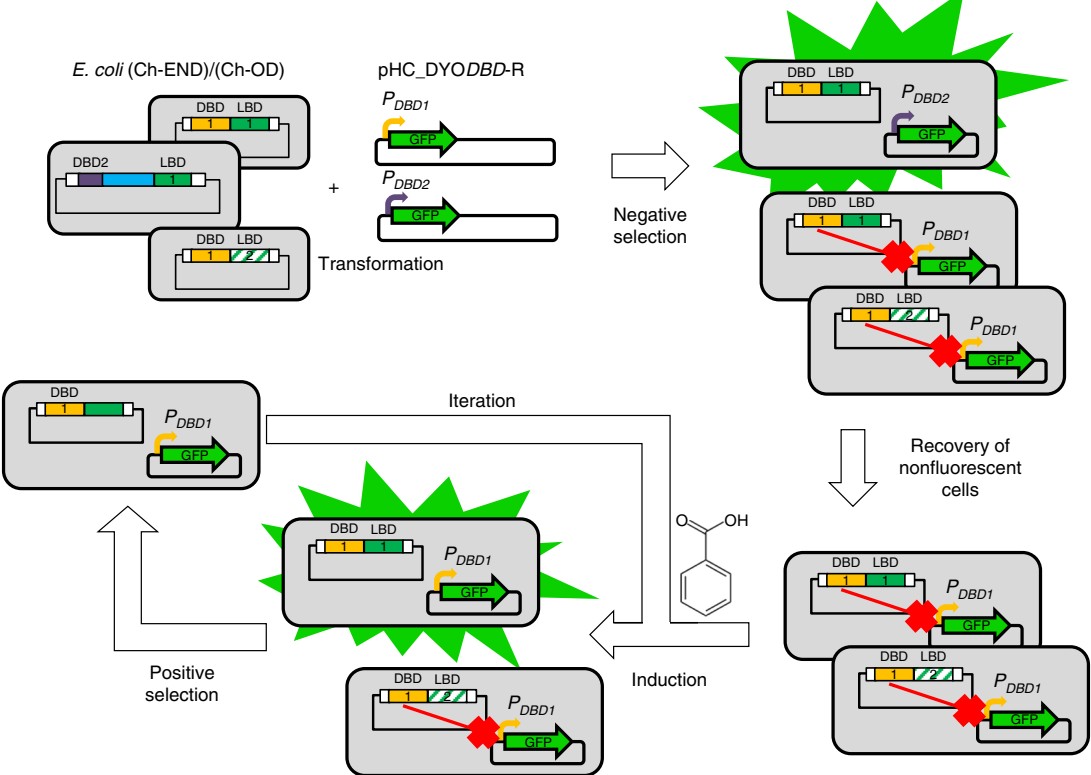

**Fig. 3** Schematic representation of enrichment process. Chemically competent *E. coli* (Ch-END)/(Ch-OD) cells (gray rectangles) carrying vectors pCKTRBS/pCKTRBS-OD expressing chimeric TFs were transformed with reporter plasmids pHC_DYO*DBD*-R as described in Methods. The resulting cells were grown in LB media in the presence of aTc. Under this culture conditions the chimeric TFs were expressed. Every TF contains one out of 15 DBD. In the cells where the DBD of the chimera (represented as yellow and purple rectangles) was able to recognize the operator boxes of the reporter promoter (bent arrows in yellow and purple) introduced in pHC_DYO*DBD*-R, and it retained its DNA-binding capabilities, the expression of GFP is repressed. On the other hand, GFP was highly transcribed in the cells where the TF was not able to interact with the reporter promoter. The cells showing the lowest levels of GFP expression were recovered using the FACS enrichment described in Methods in the so called Negative Sorting. The recovered cells were grown in the presence of the inducer molecule (benzoate). TFs carrying a functional LBD were able to recognize the inducer and transduce that information to the DBD triggering a conformational change strong enough to detach the DBD from the DNA. Under these conditions bacteria carrying functional TFs would resume the transcription of GFP from the reporter promoter. A FACS sorting allowed to obtain a population enriched in the functional chimeras (Positive Sorting). This enrichment process was iterated

**Screening of TF libraries using synthetic reporters**. Functional screening was based on two-step enrichment, as represented in Fig. 3. In the first step (Negative Sorting), the population was sorted based on GFP expression (by the reporter promoters) to enrich cells in which a pairing between the DBD of the chimera and the operator boxes had formed (GFP negative). In such cells, the chimeric TF binds to the reporter promoter and represses the expression of GFP. Accordingly, this allowed for functional screening by negative selection to recover a sorted population enriched in properly paired chimera-reporter plasmids. Bacteria containing a correct pairing of TF and reporter but where the chimeric regulator was not a functional repressor (e.g., due to erroneous protein folding or lack of secondary modifications within the polypeptide) were discarded. In a second step (Positive Sorting) bacteria recovered by negative selection were cultured in the presence of the inducer molecule benzoate, followed by sorting for positive expression of GFP. To be recovered in this step, a bacteria that expressed a functional chimera required: (a) an LBD that could bind benzoate and (b) a subsequent conformational change upon binding that could alter the allostery of the repressor, forcing it to unbind from its DNA operator within the reporter promoter. In cells where these two events occurred, RNA polymerase $\sigma^{70}$ could recognize the unrepressed $-10/-35$ boxes of the promoter and transcribe GFP, leading to cellular fluorescence. FACS sorting conditions were set to recover cells

with the highest fluorescence in the GFP emission range. This functional screening by positive selection was repeated ($n = 4-5$) to improve enrichment.

The viability of using FACS for our enrichment experiments was confirmed by control sorting experiments, as described in Supplementary Methods. Initially, we recovered a functional wild-type regulator, LacI, from a library of TFs. In a second phase we used the glucose-inducible chimera LacI-GGBP-OD (SLCP$_{GL}$) as our reference strain, as it is one of the few published chimeric transcriptional repressors[22]. LacI-GGBP-OD was considerably more abundant in our libraries after enriching for glucose-induced TFs, both when this chimera was added to the pool exogenously and when it was assembled as another fusion-gene present in the library with benzoate-binding chimeras (Supplementary Methods).

These results strongly support the applicability of our chimeragenesis process, described herein, for development of tailor-made TFs able of detecting small molecules of interest. When screening for benzoate-responsive chimeras, AYC Lib-Ch-END and AYC Lib-Ch-OD were grown separately in lysogeny broth (LB) medium supplemented with anhydrotetracycline (aTc). Nonfluorescent cells, plus the 10% of cells within the lowest fluorescence emission range, were sorted and retained. Recovered cells were grown in the presence of aTc and benzoate (benzoate-responsive TFs allowing GFP expression) and

subjected to several enrichment cycles in which the cells with the highest fluorescence were recovered and retained each time. Supplementary Table 4 summarizes the percentages of cells recovered for both AYC Lib-Ch-END and AYC Lib-Ch-OD libraries in each step of the enrichment process. The distribution of TFs in the starting libraries compared to those recovered after four or five cycles of enrichment was analyzed by next generation sequencing (Methods). No single chimera dominated the population, but on average 50 chimeras were present in the enriched libraries in an abundance equal or greater than 0.5%. Supplementary Figure 3 represents the distribution in the population for the most abundant chimeric TFs present in AYC Lib-Ch-END and AYC Lib-Ch-OD after all enrichment cycles.

These results show that in our quest to create a synthetic benzoate-responsive chimera there was no single optimal solution, but instead we created an array of chimeric TFs that were each functional to some extent. This was indeed the expected outcome of our library analysis, as the chimeras assembled here have not been subjected to the continuous forces of evolution that natural TFs have experienced.

**Validation of representative chimeric regulators**. Over 100 chimeras showed enrichment in our screening assay, and 2 of these were selected for further characterization owing to their unique features. We first chose CbnR-ABE44898-OD (ChTFBz01), from the AYC Lic-Ch-OD library, because it carried a Ct LacI OD and its LBD (ABE44898) was similar to RPA0668: a *Rhodopseudomonas palustris* protein whose ability to bind benzoate has already been established[25,40]. It is important to note that even though ABE44898 was classified as a PBP, its SP (included in this construction) sequence was the most divergent among all LBDs. This peculiarity may have had conferred interesting biophysical properties to the chimera, as the SP may also function as an extended linker sequence. We then chose LmrR-BzdB1_nSP (ChTFBz02), from the AYC Lib-Ch-END library, as it contained a LBD originally identified as a PBP associated to a cluster responsible for benzoate catabolism[41,42] The inclusion of proteins like BzdB1, associated with clusters involved in the metabolism of small soluble molecules, allowed for improvement of the libraries by addition of domains with increased probability of binding to the inducer molecule. The lack of SP in the chimera likely reduced the flexibility of the DBD–LBD interface, which may positively influence the transmission of steric changes between certain DBD–LBD combinations.

ChTFBz01 and ChTFBz02 were recloned to pCKTRBS-OD and pCKTRBS, respectively. Subsequently, *E. coli* strains harboring each were transformed with the appropriate reporter vectors. The resulting strains AYC ChTFBz01 and AYC ChTFBz02 (Supplementary Data 1) allowed us to assess the ability of CbnR-ABE44898-OD and LmrR-BzdB1_nSP to repress the expression of GFP and be derepressed by benzoate. In both cases, there was a reduction of the GFP fluorescence of the culture when the chimeras were expressed in regular culture medium (chimeras expressed, GFP promoter repressed), detailed in Methods. However, when the inducer benzoate was added to cultures, the strains showed GFP levels approximate to the basal expression of the reporter promoters. The fold difference of GFP expression between the repressed and induced conditions was $3.06 \pm 0.74$ for CbnR-ABE44898-OD and $3.13 \pm 0.31$ for LmrR-BzdB1_nSP. Figure 4 shows the GFP expression levels of tested reporter promoters when the chimeric repressors were expressed. The addition of benzoate restored GFP expression approximate to basal levels. This suggested that both CbnR-ABE44898-OD and LmrR-BzdB1_nSP were functional sensors capable of repressing their target promoters in the absence of their inducer molecule

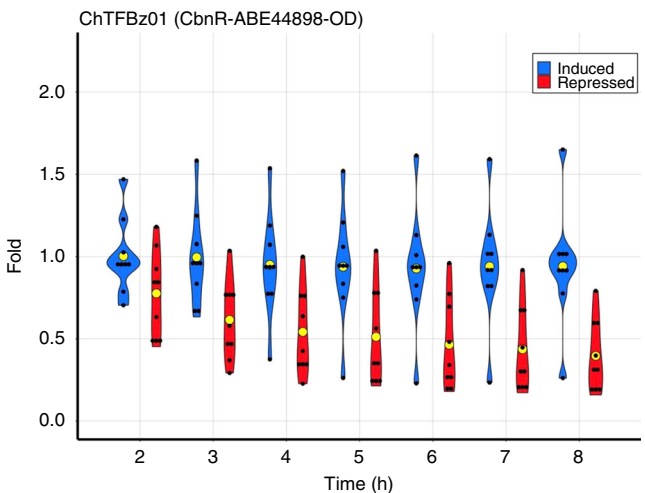

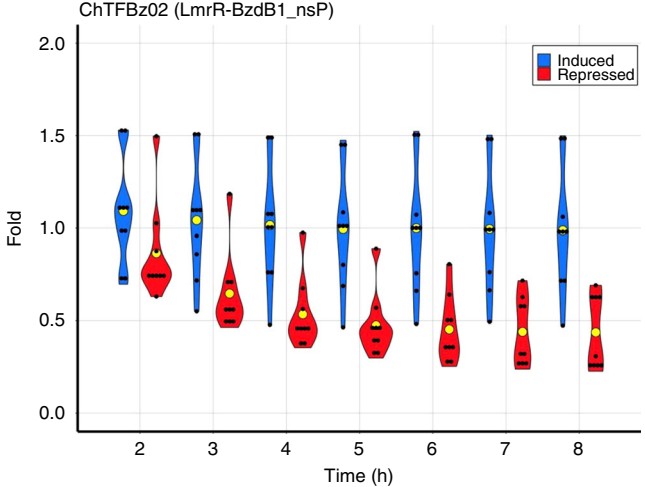

**Fig. 4** In vivo validation of ChTFBz01 (CbnR-ABE44898-OD) and ChTFBz02 (LmrR-BzdB1_nSP). Time course showing relative GFP fluorescence of AYC ChTFBz01 and AYC ChTFBz02 strains grown in minimal medium in a multiwell plate reader as indicated in Methods. Promoter activity is measured as fold of the relative fluorescence (fluorescence in arbitrary units/OD600) of the strains grown under inducing conditions (cultures supplemented with aTc and benzoate; blue violin plots) or repressing conditions (cultures supplemented with aTc; red violin plots) compared to the basal expression of the promoter (no aTc and no benzoate). The average value for every condition at a given time point corresponds to a yellow circle ($n = 6$–10). It can be appreciated how in order to maintain the basal activity (fold = 1) the addition of benzoate to the culture medium is necessary when the chimeras are expressed by the addition of aTc. The difference between repressed and induced expression is statistically significant from the third hour of the culture on. Growth conditions and fluorescence assays performed as described in Methods

benzoate, but that this repression activity was attenuated by the presence of benzoate. A detailed in vitro study of the interaction between the purified chimeras and the reporters would be needed to calculate the factors influencing the kinetic parameters of repressor-promoter and repressor-inducer interactions, but the data presented here shows these synthetic regulatory circuits are functional in vivo. It is important to note that there is also considerable opportunity to optimize these novel synthetic regulators using technologies such as directed evolution or semirational protein design[43]. Currently, the organization of these synthetic TFs is equivalent to the point in LacI's

evolutionary history where its DBD recruited an ancestor of its LBD domain[21]. That event was followed by millions of years of genetic evolution that optimized the communication between the two domains and shaped the ancestral LacI chimera into the efficient TF we know today.

The chimeragenesis method described in this study is based on the premise that most TFs resulting from the assembly would be either nonfunctional or toxic to the host cell[38]. One of the main advantages of our FACS enrichment system is that it assays the functionality of every construct in the competitive environment of a bacterial library. Toxic and nonfunctional constructs are purged from the libraries in successive enrichment steps that are tuned to increase stringency at each step. We have demonstrated the utility of this pipeline by obtaining functional biosensors for benzoate. Additionally, while we focused our efforts on obtaining benzoate-responsive TFs, our library was also designed to generate a transcriptional repressor that could be induced by 4-hydroxybenzoate. Both benzoate and 4-hydroxybenzoate are environmentally relevant, due to their association to the degradation of lignin[25,44,45]. Their abundances make them principal targets in the valorization of lignin and its associated aromatic monomers[46]. To the best of our knowledge there are no published transcriptional repressors directly induced by benzoate. The catabolism of this aromatic molecule has been thoroughly studied in anaerobic pathways. In such cases, the most frequent inducer for repressors associated with these clusters is the first metabolite of the pathway benzoyl-CoA[47], requiring the attachment of subsidiary benzoyl-CoA ligase enzymes[41,48] to the repressor-promoter system in order to apply them to synthetic regulatory circuits. Indeed, BzdR[48] and BoxR[48], two well-studied regulators associated with benzoate catabolism in anoxic conditions respond to benzoyl-CoA and not to benzoate. It is unknown whether benzoate or benzoyl-CoA are the actual inducers of one and two component systems repressing catalytic pathways related with benzoate in different microorganisms, (e.g., BamVW in *Geobacter metallireducens*[49], BadM in *R. palustris*[50], and BgeR in *Geobacter bemidjiensis*[51]). BenM, a transcriptional activator capable of recognizing benzoate as an inducer molecule has been extensively studied[30,52,53]. Despite the existence of this activator we still consider that a benzoate-responsive transcriptional repressor has added advantages.

The functionality of the TFs created in our libraries is based on the simplest transcriptional repression mechanism: interference with RNA-polymerase sigma factor binding of DNA[54]. Moreover, the chimeric genes created in this work present a single binding pocket for a small molecule, i.e., the binding site of the PBP domain, whereas BenM can simultaneously bind two inducers (benzoate and *cis,cis*-muconate)[29] leading to more complicated inducer landscapes and potentially nonspecific activations that are unrelated to benzoate.

In this study, we describe a comprehensive strategy to create custom monogenic biosensors using fusion of modular components. Using this strategy, we could independently replicate the construction of a previously characterized chimeric glucose-sensing TF (SLCP$_{GL}$[22]) as well as construct multiple novel benzoate-sensing TFs and further characterize two of them: ChTFBz01 and ChTFBz02.

The novel sensor proteins developed in this work expand the limited collection of available transcriptional repressors that can be used as biosensors in the degradation of lignin. Beyond their immediate usability to tackle this biotechnological problem, they represent an important milestone for the construction of synthetic TFs on demand, as our pipeline can be readily applied to create many other custom-made chimeric TFs. The use of PBPs as detection domains underscores the potential of this method for generating tailored biosensors, and

goes far beyond the swapping of domains between currently known regulators.

This pipeline represents an enabling first step toward construction of synthetic regulatory circuits. Nevertheless, to achieve their full potential when addressing real-life biotechnological problems, new TFs will greatly benefit from further progress in fine tuning of gene expression. It is well-known that the expression of any transgene, and in particular TFs, brings with it a potentially deleterious metabolic burden for the host cell[38]. Fortunately, recent publications have increased our understanding of the delicate underpinnings that permit fine-tuning of gene expression[15,16,55]. Owing to these advances, we can modulate dynamic range, threshold, and ligand affinity of TFs without first resorting to protein engineering (e.g., modifying promoter strength, operator boxes, and RBS). In this context novel strategies, such as the one presented here, can focus more on obtaining TFs capable of recognizing new ligands and less on improving their ligand affinities and dynamic ranges. Dedication to the construction of new biosensors is especially relevant in the Synthetic Biology field, given the limited availability of bio sensing modules[15]. Conversely, as new TFs are constructed, there will be an increase in number of protein chassis available to facilitate the engineering of binding pockets for new ligands[56].

Finally, the systematic construction of many fusion genes by assembling different collections of modules in a logical order was made possible by the development of eLCR: a novel strategy to assemble highly diverse fusion libraries presented in this work. While reserving the capacity to attach DNA fragments in a scarless fashion, eLCR also enables incorporation of short-DNA sequences between said fragments. This characteristic makes this procedure ideal for constructing mutant protein libraries that carry, for example, linkers of different lengths or clusters of mutations. Another strength of this method resides in its potential to create novel TFs through straightforward design, but does not demand advanced knowledge of protein structure to succeed.

In summary, the chimeragenesis method described in this work facilitates the creation of, to the best of our knowledge, new biosensors and, coupled with improvements in tunability of gene expression, has the potential to become an enabling standard for researchers seeking to construct synthetic biosensor circuits.

## Methods

**Bacterial strains and growth conditions**. Bacterial strains and plasmids used are listed in Supplementary Data 1. *E. coli* strains were grown at 37 °C in LB medium[57]. When required, *E. coli* cells were grown in M63 minimal medium[58] using the necessary nutritional supplements and 30 mM glycerol (Sigma, St. Louis, MO) as a carbon source. Antibiotics were added at the following concentrations: 100 μg ml$^{-1}$ ampicillin/carbenicillin and 25 μg ml$^{-1}$ chloramphenicol (Sigma, St. Louis, MO). For protein expression from pCKTRBS/pCKTRBS-OD plasmids, cultures were induced with 0.5 μg ml$^{-1}$ anhydrotetracycline (aTc) (Clontech, Mountain View, CA). When experimental conditions required, 1 mM IPTG (Sigma, St. Louis, MO), 1 mM sodium benzoate (Sigma, St. Louis, MO) and 0.4% glucose (Teknova, Hollister, CA) were supplemented.

**Molecular biology techniques**. Molecular biology techniques were performed following commonly used standard protocols and as per manufacturers' instructions[59]. Plasmid DNA was purified with a Qiaprep Spin Miniprep Kit (Qiagen, Hilden, Germany). DNA fragments were purified with DNA Clean-up and Concentration Kit (Zymo research, Irvine, CA). The oligonucleotides employed for PCR amplification of the cloned fragments and other molecular biology techniques are summarized in Supplementary Data 2 and were supplied by IDT (Coralville, IA). All cloned inserts and DNA fragments were confirmed by Sanger sequencing[60] performed by Genewiz Inc. (Cambridge, MA). Commercially available *E. coli* NEB5-alpha and *E. coli* NEB5-alpha F' I$^q$ chemically competent cells (NEB, Ipswich, MA) were used for routine transformations. Alternatively, electrocompetent *E. coli* cells were generated and transformed immediately (Gene Pulser; Bio-Rad, Hercules, CA)[59]. Cloning was routinely performed by Gibson assembly[61]. Nucleotide sequence analyses were done at the National Center for Biotechnology Information server (www.ncbi.nlm.nih.gov). Prediction of SPs was performed

using SignalP[62] at the Technical University of Denmark online server (http://www.cbs.dtu.dk/services/SignalP/).

**Construction of expression vectors pCKTRBS and pCKTRBS-OD.** pCK01[63] was the chassis of choice for the construction of plasmid pCKTRBS. This vector was tasked with the expression of all the chimeric TFs generated in this work. The *Plac* promoter on pCK01 was replaced by a *tetR-PtetO* cassette modified from the plasmid pKD154 (Wanner and Datsenko, unpublished). *PtetO* was the inducible promoter transcribing the chimeras. In the absence of aTc *PtetO* was repressed by TetR and the chimera did not get expressed, whereas in the presence of aTc TetR unbound *PtetO*, opening the system and enabling transcription from it. The *tetR-PtetO* carrying pCK01 derivative was furthermore modified to include a consensus RBS sequence downstream of *PtetO*, hence the name pCKTRBS (T for *tetR*, RBS for the consensus Shine-Dalgarno). RBS and the adjacent upstream bases constituted a 5′-homology arm included in all the DBD domains. A fragment of the old pCK01 polylinker located at 3′ of the newly inserted RBS was used as the homology arm at 3′-end of the LBDs and amplified to be cloned without further modification.

A variant of pCKTRBS containing LacI OD was constructed introducing OD between the consensus RBS and the remains of the former pCK01 polylinker. The amplification of the chimeras with a 3′-arm homolog to LacI OD granted the cloning of the TFs as a translational fusion to OD in their Ct end. This new vector was named pCKTRBS-OD. Since TetR was constitutively transcribed, the expression of the genes cloned in pCKTRBS/pCKTRBS-OD was repressed by default. In the cases where it was necessary to induce the expression of a cloned gene, aTc was added to the culture media as indicated above. A detailed representation of the region comprising from *tetR* to OD in pCKTRBS-OD including operator boxes for TetR and the RNA polymerase σ[70] subunit located on the expression promoter *PtetO* is shown in Supplementary Fig. 2. The fully annotated sequence of pCKTRBS and pCKTRBS-OD is shown in Supplementary Methods.

**Preparation of dsDNA and ssDNA fragments for eLCR.** DBD-CORE and LBD domains were synthesized as dsDNA fragments by the custom DNA manufacturer Gen9 (Cambridge, MA). Silent mutations were incorporated as requested by the manufacturer when it was necessary to remove target sites for restriction enzymes that would be needed for the assembly process.

All possible DBD-CORE-ENDS-(LNK)-LBD sequence combinations, as well as a list containing all oligonucleotides necessary to assemble them, were obtained using a custom Perl script. Two types of oligonucleotides were employed in the assembly depending on their orientation and the DNA strand they would conform when ligated. Those identified as "Infra" (Infra from now on) had the same sequence and orientation as the template strand, whereas those labeled as "Supra" (Supra from now on) carried the same sequence and orientation as the coding strand (Fig. 2). Specific adapters for Infra and Supra were engineered based on two orthogonal pairs of primers who had been designed to selectively amplify pools of oligonucleotides from high-fidelity DNA microchips[64]. These adapters were incorporated into the oligonucleotide synthesis order as 5′ and 3′- flanking sequences (Supplementary Fig. 4).

Three Custom Array chips were used to synthesize the 223170 required oligonucleotides. DNA synthesis in microarrays was performed by Daniel Wiegand in the Synthetic Biology Platform from the Wyss Institute at Harvard (Boston, MA). Oligonucleotide libraries were synthesized using standard phosphoramidite chemistry on a 90K CustomArray DNA Oligonucleotide Library Synthesis microarray using the B3 Synthesizer platform. After libraries were synthesized, the surface bound oligonucleotides were cleaved from the microarray by incubation in 30% ammonium hydroxide at 65 °C for 12 h. Oligonucleotide libraries were then dried and concentrated with a SpeedVac Concentrator set at 65 °C and vacuum engaged for 3 h. The resulting pellet was resuspended in 70 µl of 1× TE buffer and purified using a P-30 size exclusion column (Biorad) pre-equilibrated with 1× TE buffer. The concentration of the resulting purified oligonucleotide libraries was determined with a NanoDrop spectrophotometer at A260. The ssDNA pool resulting from the synthesis was amplified to preserve the original sample and to enable the obtention of a more concentrated suspension of oligonucleotides.

Supplementary Figure 4 is a schematic representation of adapters enabling the amplification of the oligonucleotide library (Panel A) as well as of the purification process (Panel B). The pool of oligonucleotides obtained from the microarray synthesis was PCR amplified in two different reactions. The primer pair Lib_Adapt_Supra_5/Lib_Adapt_Supra_3 amplified specifically Supra ssDNA, whereas Lib_Adapt_Infra_5/Lib_Adapt_Infra_3 amplified Infra oligonucleotides. Lib_Adapt_Supra_5 and Lib_Adapt_Infra_5 oligonucleotides used for the PCR reaction carried two modifications: (i) biotin at the 5′-end and (ii) uracil in place of a thymidine at the 3′-end. These modified versions of the oligonucleotides were identified as BioT-Lib_Adapt_Supra_5-U and BioT-Lib_Adapt_Infra_5-U. PCR amplification of the oligonucleotide libraries using the modification-bearing oligonucleotides as primers generated biotinylated and uracilated dsDNA fragments. Interestingly, the amplification yield of Infra was consistently lower than that of Supra (Infra was on average 77.3 ± 13.4% of Supra), especially when the templates were complex libraries; suggesting this was not a result of inferior performance by the primer set, but instead a more intrinsically difficult template library.

Once the CustomArray DNA pools had been amplified it was necessary to remove the adapter sequences that enabled the hybridization of Lib_Adapt_Supra_5/Lib_Adapt_Supra_3 and Lib_Adapt_Infra_5/Lib_Adapt_Infra_3. Type IIS restriction enzymes were used in a first step to remove nonbiotinylated ends, as they cut dsDNA bases away from their recognition motifs in target DNA. The Lib_Adapt_Supra_3 and Lib_Adapt_Infra_3 adapters incorporated BsmFI and BspQI target sequences, respectively. dsDNA Supra fragments were digested with BsmFI and Infra fragments with BspQI, following manufacturers specifications (NEB, Ipswich, MA). Digested pools were incubated with Dynabeads M-280 Streptavidin (Life Technologies, Carslbad, CA). The dsDNA oligonucleotides were attached to the paramagnetic beads through the biotin and the adapters that were digested were washed and discarded. Nonbiotinylated strands were not directly attached to the beads and could be removed by an alkaline denaturation in 125 mM NaOH at 90 °C for 2 min, washed and discarded. The Supra and Infra oligonucleotides were detached from the beads and removed from the remaining adapter through USER digestion (NEB, Ipswich, MA), that cut the DNA in the uracil incorporated at the 3′-end of the biotinylated primers.

The final ssDNA Supra and Infra pools were purified using DNA Clean-up and Concentration Kit (Zymo, Irvine, CA). Given the average proportion between Supra and Infra oligonucleotides and their respective flanking adapters, as well as the fact that dsDNA enters the Type IIS digestion–denaturalization–USER digestion process and ssDNA is retrieved, the maximum possible yield is close to 30% in terms of mass, while we experimentally observed an average 5.1 ± 2.2%. This sixfold loss can be attributed to the different purification steps. The amplified material was enough for the correct performance of the assembly reactions described below.

**Enhanced LCR.** Canonical LCR is based on the following principle: short-ssDNA fragments with complementarity to two individual dsDNA molecules act as staples to connect both fragments and allow for a DNA ligase to bind them[65]. An initial denaturation at high temperature separates the two strands of the target dsDNA fragments. When the temperature decreases the staple oligonucleotide anneals to the target sequences and the nick between the two DNA molecules is closed by a thermostable DNA ligase in a scarless process. This assembled strand serves as template for the assembly of the complementary strand in a subsequent cycle of denaturalization, annealing, and ligation. Temperature cycling allows the assembly of multiple DNA fragments in a highly efficient manner. We confronted significant challenges in attempts to apply this original LCR method to the assembly of fusion genes presented in this work. Different sequences needed to be introduced between DBD and LBD, bridging them, on top of the scarless assembly provided by LCR. In between 15 DBD-COREs and the 30 LBDs (inclusive of versions with and without SP), our work required us to add specific ENDS corresponding to each DBD-CORE and the 19 different LNK options. The length of the sequences introduced between the DBD-CORE and the LBD ranged from 3 bp (shorter DBD-END equivalent to one codon) and 243 bp (combining the longest DBD-END, 120 bp, and the longest LNK, 123 bp). We created four different classes of chimeras based on the number and complexity of the stapling oligonucleotides needed to assemble them, which was related to the length of the sequence introduced in between DBD-CORE and LBD. Class I chimeras (1 Infra) were TFs in which there was a direct connection between DBD and LBD. More than half of the library belonged to Classes II (1 long Infra, 1 Supra) and III (1 short Infra, 1 Supra), chimeras in which 3–66 bp were introduced between DBD and LBD. Finally, Class IV (2 Infra, 3 Supra) chimeras carried a >66 bp insertion. More information regarding the different classes of chimeric TFs can be found in Supplementary Methods.

The first step for the enhanced LCR (eLCR) assembly consisted of the phosphorylation of all the DNA molecules in the reaction[39]. An equimolar mix of 15 DBD-CORE and 30 LBD was mixed with amplicons of Supra and Infra oligonucleotides and phosphorylated using T4 polynucleotide kinase (Enzymatics, Beverly, MA) for 30 min at 37 °C. These phosphorylated products formed the substrates for assembly of chimeric TFs using a thermostable DNA ligase.

Before assembling the complete collection of TFs, we verified a mock library of four evenly distributed chimeras, each belonging to one of the aforementioned classes. DBDs and LBDs of this testing library were taken from the collection of DNA fragments used for the final construction of the chimeras, whereas the subset of oligonucleotides necessary to assemble them was obtained from IDT (Coralville, IA). This mock library enabled the testing of three different thermostable ligases: Ampligase (Epicentre, Madison, WI), 9°N DNA ligase (NEB, Ipswich, MA) and Taq DNA ligase (NEB, Ipswich, MA). After this preliminary optimization Taq DNA ligase appeared to be the most efficient ligase for application within the eLCR method. Another parameter that required optimization with the mock library was the thermal conditions of the ligation reaction. The only chimeras generated through a canonical LCR[39] were Class I. Temperature cycling allows a ligated strand to serve as a template for the single ligation required to generate its reverse-complementary strand, as is the case for Class I chimeras. The assembly of Classes II–IV, however, involved more complex reactions as several oligonucleotides needed to be hybridized and ligated to reconstitute the DNA strands. We found that the cycling of temperatures helped to increase the proportion of Classes II–IV in the pool. The relative distribution of the different classes of chimeras in the mock library was influenced by the cycling program of choice.

The final thermal conditions for the assembly are as follows:[66] an initial incubation at 94 °C for 2 min followed by 10 cycles of alternating 94 °C for 30 s and 45 °C for 4 min. To facilitate correct hybridization of the staple oligonucleotides, the temperature ramp decrease from 94 to 45 °C was reduced to 30% of the top speed in the Mastercycler Pro (Eppendorf, Hamburg, Germany) thermocycler used. The products of the assembly were PCR amplified with primers that hybridized with the flanking sequences introduced to the 5′ of the DBDs and 3′ of LBDs. The primer RBS_Tail_F1 hybridized in the 5′ 30 bp region common to all the DBD amplicons. The primers END_Tail_R1 and OD_Tail_R1 hybridized in the 30 bp regions designed to clone the chimeras without or with a translational fusion to OD, respectively (Supplementary Data 2). The amplification was monitored in real time using a LightCycler 96 System (Roche, Basel, Switzerland) and stopped in midexponential phase of the amplification curve, minimizing nonquantitative effects in the plateau of the reaction that would reduce diversity within the library[67].

**Cloning of eLCR-assembled genes**. Preliminary data suggested that an intermediate cloning of the amplified library using a Zero Blunt TOPO PCR cloning Kit vector (Invitrogen, Carlsbad, CA) instead of direct cloning to pCKTRBS/pCKTRBS-OD was more efficient. The purified products of the PCR amplification of the libraries, with and without OD, were cloned into pCR-BluntII-TOPO following manufacturer's instructions and transformed into chemically competent NEB5-alpha F′ I$^q$ cells (NEB, Ipswich, MA). An aliquot of the recovered cells was stored at −80 °C to preserve the maximum possible diversity of the TF collection, while another aliquot was used to isolate plasmids. These plasmid libraries were used as templates for PCR amplifications of the chimeras (monitored in real time as described above). One amplification batch included all the chimeras carrying OD (using the primer pair RBS_Tail_F1/OD_Tail_R1) and another batch included the TFs without OD (using the primer pair RBS_Tail_F1/END_Tail_F1). The amplified libraries were cloned into pCKTRBS/pCKTRBS-OD vectors using Gibson assembly. Linear pCKTRBS/pCKTRBS-OD vectors were obtained by divergent PCR amplification using the primer pairs pCKTRBS_RBS_Tail_R1/pCKTRBS_END_Tail_F1 and pCKTRBS_RBS_Tail_R1/pCKTRBS-OD_OD_Tail F1, respectively (Supplementary Data 2). The 30 bp 5′ and 3′-terminal regions of both plasmids showed homology to the sequences flanking the assembled chimeras, to facilitate integration downstream of the PtetO promoter and the RBS. The products of the pCKTRBS/pCKTRBS-OD library assemblies were transformed into chemically competent NEB5-alpha cells (NEB, Ipswich, MA). The resulting collections of bacterial cells carrying chimeric TFs cloned into pCKTRBS/pCKTRBS-OD were designated E. coli (Ch-END) and E. coli (Ch-OD), respectively (Supplementary Data 2).

**Construction of the pHC_DYODBD-R library of reporter vectors**. A set of reporter vectors designed to monitor the activity of TF libraries was constructed on a pUC19 chassis[68]. Preliminary data revealed that the expression of GFP from high-copy plasmids was more suited to monitor the activity of the TFs using FACS compared to the use of low-copy vectors. Plasmid pHC_DYOLacI-R (Supplementary Data 1) was the first reporter plasmid constructed and tested. The full sequence of pHC_DYOLacI-R and the origin of the different regions that were PCR amplified and Gibson-assembled to build it are detailed in Supplementary Methods.

pHC_DYOLacI-R carried a Ptac promoter driving the expression of the reporter gene GFP. Ptac was the scaffold on which the synthetic promoters were built upon (Supplementary Data 3). A prerequisite for selection of transcriptional repressors donating their DBDs to the chimeric TFs was that they contained empirically tested operator boxes (Supplementary Table 1). This enabled their incorporation into synthetic promoters driving the expression of the reporter genes for TF screening upon binding. Ptac was modified to include the operator boxes for the rest of the DBDs, preserving the integrity of the −10/−35 boxes to allow them to be recognized by the RNA polymerase σ[70] subunit. The distance between −10/−35 was also maintained constant for this reason. The operator boxes published for every regulator were strategically incorporated to Ptac so that transcription from the modified promoter would cease when their cognate protein partner bound to the DNA. For this purpose, the spaces between −10/−35 and between the transcription start site and −10 were preferred. In some occasions mutations in −10/−35 were required to accommodate the operators. Supplementary Data 3 contains the sequences of each synthetic promoter.

The method of choice to construct the pHC_DYODBD-R vectors consisted of divergent PCR using pHC_DYOLacI-R as template (see oligonucleotides in Supplementary Data 2). The divergent oligonucleotides amplified the whole extent of pHC_DYOLacI-R except the Ptac promoter and LacI operator boxes. There was an individual pair of primers for each reporter plasmid that was constructed. The modified Ptac sequence for each synthetic promoter was split between the 5′ (phosphorylated) and 3′ oligonucleotides. PCR products were circularized to close themselves, restoring the sequence of the new synthetic promoters. The resulting plasmids were transformed into NEB5-alpha F′ I$^q$ for propagation and storage. New constructs were identified as pHC_DYODBD-R, where DBD was substituted by the name of the regulator whose operator boxes have been introduced in the promoter (Supplementary Data 1).

NEB5-alpha F′ I$^q$ (pHC_DYODBD-R) strains were used in a two-pronged approach to validate and compare the functionality of synthetic promoters driving the transcription of the GFP-encoding reporter gene. In the first approach, we assessed the relative activity (Arbitrary fluorescence units/OD600) of individual promoters compared to Ptac. This set of experiments determined that the synthetic promoters were working within a range of one order of magnitude from the most to the least active (Supplementary Fig. 2). Precultures of the 15 strains were individually grown overnight at 37 °C in LB. The next day they were used to inoculate fresh LB cultures in a 96-well plate (Flat bottom Polystyrene Black with clear bottom; Corning, Corning, NY) and incubated at 37 °C in a Synergy H4 Hybrid Multi-Mode Microplate Reader (Biotek, Winooski, VT), where OD600 and fluorescence in the range of GFP (excitation 485 nm, emission 528 nm) were monitored along a time course of 12–24 h. NEB5-alpha F′ I$^q$ (pHC_DYOLacI-R) was included in every plate as reference strain, supplemented with 1 mM IPTG.

In the second approach, 15 different NEB5-alpha F′ I$^q$ (pHC_DYODBD-R) strains were grown in separately using the conditions described above or inoculated and grown together. Fresh 3-h old cultures containing a mix of strains, carrying the reporter plasmids, were cell sorted using an Avalon Cell Sorter (PROPEL labs, Fort Collins, CO). Cells with the strongest fluorescent emission signal for GFP (approximately 3–5% of the total) were recovered and plated on LB agarose plates containing carbenicillin. Individual clones were selected and the reporter promoters were identified by Sanger sequencing. The promoter abundance in the presorting library was also characterized through Sanger sequencing of individual clones. The relative distribution of the promoters in the library before and after the selection process was not statistically different when Ptac was removed from the data set.

**Transformation of reporter plasmids into E. coli Ch-END/-OD**. Once the library of chimeric TFs was cloned into pCKTRBS/pCKTRBS-OD and the collection of reporter promoters were integrated into the pHC_DYODBD-R vectors, it was necessary to obtain E. coli cells carrying one plasmid of each kind together. The screening method implemented in this work was based in a two-step FACS enrichment, with no selection marker pairing the DBD of a given chimera with its tailor-made reporter promoter. Both pCKTRBS/pCKTRBS-OD and pHC_DYODBD-R carried antibiotic resistance markers, Cm$^R$ and Ap$^R$, respectively, but these genes were exclusively involved in plasmid maintenance within the host strain. As the copy number of the pUC19 based pHC_DYODBD-R vectors was higher than that of the pCK01 based pCKTRBS/pCKTRBS-OD, the former needed to be transformed into E. coli (Ch-END)/(Ch-OD) cells carrying the latter. The maximum possible efficiency was required for the transformation to increase the probability of generating a population of cells in which chimera and reporter were correctly paired. Electroporation of the reporter plasmids into the chimera libraries returned consistently low efficiencies ($10^5$ cfu/μg). To compensate for this, we instead utilized the RbCl procedure[59] for preparation of competent cells, which increased the efficiency upwards of $10^8$–$10^9$ cfu/μg. Three aliquots of competent E. coli (Ch-END)/(Ch-OD) cells were independently transformed with 100 ng of each pHC_DYODBD-R plasmid resulting in an average $8.0 \pm 2.9$-fold coverage of the library per individual reporter plasmid. Transformed cells were pooled and selected in liquid media supplemented with chloramphenicol and carbenicillin. After 7 h of liquid selection the bacteria were stocked at −80 °C. The two new libraries obtained after the recovery were identified as AYC Lib-Ch-END, short name for NEB5-alpha (pCKTRBS-Chimera, pHC_DYODBD-R), and AYC Lib-Ch-OD, moniker for NEB5-alpha (pCKTRBS-Chimera-OD, pHC_DYODBD-R), where Chimera represents any Nt-DBD-(LNK)-LBD-Ct.

**Relative abundance of chimeric TFs in bacterial populations**. Sanger sequencing[60] of the pCKTRBS/pCKTRBS-OD vectors using the oligonucleotides pCKPolyF1 and pCKPolyR1 (Supplementary Data 2) was used in test experiments where the relative abundance of a single chimeric TF compared to the rest of the library was required, but without a deep profile of the library. Presorting and postsorting samples were streaked to LB plates supplemented with the appropriate antibiotics and incubated at 37 °C to obtain individual clones. A random set of clones was submitted to Genewiz Inc. (Cambridge, MA) for plasmid sequencing.

In some cases, a more concise estimate of the chimeric TF profile in pCKTRBS/pCKTRBS-OD was required. Here, plasmids in the libraries were purified using commercially available kits (Qiagen, Hilden, Germany). These plasmid libraries served as template DNA for amplification of the joint region connecting the different domains of the chimeric TF. For each chimera, a region spanning the DBD-3′ to 5′-LBD was PCR amplified using specific oligonucleotides (Supplementary Data 2). The Forward oligonucleotides hybridized at positions close to the 5′-end of the DBD-CORE and the reverse oligonucleotides hybridized upstream of the SP region of the LBD. The resulting PCR products acted as barcodes containing sufficient information to identify DBD-CORE, DBD-END, LNK (if present), LBD-SP (if present) and LBD. To minimize PCR amplicon size bias, the reaction was monitored in real time as described above. Both the forward and reverse oligonucleotides contained flanking sequences enabling barcoding at both ends for pool paired-end sequencing performed in the MiSeq platform (Illumina, San Diego, CA). The resulting reads were analyzed using the DNA alignment program Bowtie[69]. Subsequent handling of the data was performed with R (https://www.r-project.org/) in the RStudio environment (RStudio Inc., Boston,

MA). Supplementary Table 3 contains a summary of the estimated distribution of the chimeric genes cloned in the library in the different Class types described in Fig. 2 and Supplementary Methods.

**FACS-based enrichment of ligand-binding chimeric TFs.** Fresh cultures of *E. coli* AYC Lib-Ch-END and AYC Lib-Ch-OD were inoculated from overnight pre-cultures. Bacteria were grown in LB supplemented with chloramphenicol, carbenicillin, and aTc. When the induction of the control chimera LacI-GGBP-OD was required the culture was supplemented with 0.4% glucose. In the screening for benzoate-binding TFs the cultures were supplemented with 1 mM benzoate. After growing for 4 h, 2 ml of the cultures were precipitated (1 min at 20,600×g in a standard benchtop microcentrifuge). Supernatants were discarded and pellets resuspended in 1 ml PBS pH 7.2. Samples were diluted in PBS pH 7.2 at a density of $10^7$ cells ml$^{-1}$ and sorted on a SH800Z Cell Sorter (Sony Biotechnology, San Jose, CA) using a 100 μM sterile sorting chip. The gating strategy (Supplementary Fig. 5) consisted of a population level round gate on a bivariate plot of forward scatter (FSC) area vs. back scatter area, followed by a rectangular gate containing various percentages of enhanced GFP-Positive cells (EGFP$^+$) on a bivariate plot of FSC area vs. EGFP area. Excitation laser emission was 488 nm. Emission was collected with a 525/50 band pass filter. The sort mode was "Ultra Purity" and the event rate was approximately 5000 per second. Cells were sorted into PBS pH 7.2 buffer. The optical layout of the sorter can be found in the Operator's Guide (Sony Biotechnology, San Jose, CA).

**Cloning of the benzoate-responsive TFs ChTFBz01 and ChTFBz02.** The ChTFBz01 (CbnR-ABE44898-OD) gene (Supplementary Methods) was ordered split on two synthetic dsDNA fragments (IDT, Coralville, IA). The first of them was flanked in its 5′-end by a 30-bp consensus RBS sequence included in pCKTRBS-OD (5′-CGGTACCCGGGTGACCTAAGGAGGTAAATA-3′) and PCR amplified with primer pair RBS-Tail_F1/cbnR_core-ABE44898-OD R1. The second fragment was flanked in its 3′-end by a 30-bp homology arm to the OD domain included in pCKTRBS-OD (5′-AAAAGAAAAACCACCCTGGCGGCCCAATACG-3′) and PCR amplified with primer pair cbnR_core-ABE44898-OD F1/END-OD-Tail_R1. A 30 bp homology region present in both the 3′-end of the first fragment and the 5′-end of the second fragment allowed for correct cloning of the chimeric gene through a three-way Gibson assembly with a linear pCKTRBS-OD plasmid obtained by divergent PCR amplification (pCKTRBS RBS-Tail R1/pCKTRBS-OD END-OD Tail F1). The assembled product was designated pCKTRBS-CbnR-ABE44898-OD and transformed to chemically competent NEB5-alpha cells (NEB, Ipswich, MA). After chimeric construct verification by Sanger sequencing, the reporter plasmid for DBD-CbnR was electroporated to the validated chimera-bearing clone giving place to the NEB5-alpha (pCKTRBS-CbnR-ABE44898-OD and pHCDYOCbnR-R) strain. This strain was designated AYC ChBzTF01 (Supplementary Data 1).

The ChTFBz02 (LmrR-BzdB1_nSP) gene (Supplementary Methods) was ordered as a single synthetic dsDNA fragment (IDT, Coralville, IA) flanked in its 5′-end by the 30 bp consensus RBS sequence mentioned above and in its 3′-end by a 30 bp sequence present on pCKTRBS polylinker (5′-GATCCTCTAGAGTGGACCTGCAGGCATGCA -3′). The fragment was PCR amplified using the primer pair RBS-Tail_F1/END-Tail_R1 and Gibson assembled into a linear pCKTRBS vector obtained by divergent PCR using the primer pair pCKTRBS RBS-Tail R1/pCKTRBS END-Tail F1. The assembled product was named pCKTRBS-LmrR-BzdB1_nSP and transformed into chemically competent NEB5-alpha cells (NEB, Ipswich, MA). Once the chimeric construction was verified by Sanger sequencing the reporter plasmid for DBD-LmrR was electroporated to the validated chimera-bearing clone giving place to the NEB5-alpha (pCKTRBS-LmrR-BzdB1_nSP, pHC_DYOLmrR-R). This strain was designated AYC ChBzTF02 (Supplementary Data 1).

**In vivo assay of ChTFBz01 and ChTFBz02 activity.** AYC ChTFBz01 and AYC ChTFBz02 strains were used to assay the in vivo functionality of the new chimeric repressors. Precultures of the two strains were individually grown overnight in LB medium. These were then used to inoculate fresh M63 minimal medium cultures in a 96-well plate (Flat bottom Polystyrene Black with clear bottom; Corning, Corning, NY) that was in turn incubated at 37 °C in a Synergy H4 Hybrid Multi-Mode Microplate Reader (Biotek, Winooski, VT), where OD600 and fluorescence in the emission range of GFP were monitored along a time course for a minimum of 8 h.

**Code availability.** The code that support the findings of this study is available from the corresponding author upon reasonable request.

**Data availability.** The data that support the findings of this study are available from the corresponding author upon reasonable request.

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

## Acknowledgments

This material is based upon work supported by the U.S. Department of Energy (DOE), Office of Science, Biological and Environmental Research Program under Award number DE-FG02-02ER63445 (PI G.M.C.), which has sponsored J.F.J., B.L.A., S.L.B., and G.M.C. J.F.J. and C.D.J. are supported by the National Institute Of Dental and Craniofacial Research of the National Institutes of Health (NIH) under Award number R01DE027850 (PI C.D.J.). This work is solely the responsibility of the authors and does not necessarily represent the official views of the DOE or the NIH. We want to acknowledge H.H. Lee, R. Kalhor, N. Ostrov, and A.H. NG (Harvard Medical School-HMS) as well as I.F. Escapa (The Forsyth Institute) for their help discussing and reviewing this work. We thank F.J. Carrillo-Salinas (Tufts University) for the graphic design of Fig. 2, D.J. Wiegand and B. Turczyk (Wyss Institute) for the CustomArray DNA synthesis, J.K. Moore and S. Terrizzi (HMS) for their assistance with FACS, N. Eroshenko (HMS) for his oligonucleotide collection and B.L. Wanner (HMS) for sharing his vectors. J.F.J. thanks J. Aach (HMS) and V. Raman (University of Wisconsin-Madison) for the seminal discussions that originated this paper and E. Díaz, G. Durante-Rodríguez, and M. Carmona (CIB-CSIC) for introducing him to chimeric TFs.

## Author contributions

J.F.J. and G.M.C. developed the concepts and conceived of the study plan. G.M.C provided project administration, supervision, and resources. J.F.J. designed the experiments and performed them with assistance from B.L.A. and S.L.B. for TF screenings. J.F.J. analyzed the data and drafted the manuscript with contributions from G.M.C. C.D.J. provided resources and critically rewrote the manuscript with contributions from J.F.J.

## Additional information

**Competing interests:** The authors declare no competing interests.

