## [Peer Review File · Nature Communications]

Reviewers' comments:

Reviewer #1 (Remarks to the Author):

This manuscript reports results of a protein engineering and screening project to create artificial transcription regulators. The strategy was to join protein domains that do not naturally occur together, using a massive recombination library to vary the two functional domains and connecting linkers, along with an oligomerization domain. A screening pipeline was developed to identify which novel proteins were capable of repressing transcription and responding to small molecule inducer signals.

This manuscript summarizes a great amount of work. The main result was the creation of a novel recombination pipeline and cloning method (eLCR) that enabled enhanced variation in the protein sequences to be screened. Upon screening >130,000 novel protein sequences, the authors do identify two novel chimeras that have a measurable induction to an uncommon inducer (benzoate). As they acknowledge, there is room for improvement in the protein function: Induction is ~3 fold for the two constructs identified; the best natural repressors respond up to 1000-fold (although many are probably in the range of 10-100 fold). The authors also mention the need to biochemically characterize the two repressors as well as to demonstrate that benzoate (and not some metabolite) is the actual inducer.

The authors are to be commended for their thorough details in the methods.

Although it would be hard to switch terms mid-way through a project, I think the authors should reconsider the name of "substrate" binding domain (SBD). To me, "substrate" has a very specific definition of a ligand undergoes enzymatic catalysis; this process does not occur in these transcription regulators. Alternative names commonly used for these types of proteins include "ligand" binding domains (although DNA is also a ligand, so it isn't perfect) and "regulatory" domains (because small molecule binding to this domain regulates DNA binding of the other domain).

In several places, paragraphs become very long and could be split (for example on page 11 and in Methods).

Line 628 contains a possible typo in the phrase "in a subsequent cycle of denaturalization". Should the last word be "denaturation"?

Reviewer #2 (Remarks to the Author):

This paper presents the construction of chimeric transcription factors that respond to specific metabolites. Metabolite biosensors are an important challenge in metabolic engineering and subject of much research in the field at the moment. This is a well written paper with two important contributions: 1) the protein engineering pipeline to build and select for the target function, 2) a benzoate-responsive repressor that can be used in applications.

I missed more discussions on the tunability of the biosensors and their expected dose-response curve - the benzoate sensor was tested for a single concentration only. So far most metabolic engineering applications have focused on metabolite-responsive TFs to sense intra or extracellular metabolites. The dose-response curve of these can be rapidly tuned with edits to the promoter sequence or the expression level of the TF. Can the same level of tunability be achieved with the chimeric TFs proposed in this paper? The authors should discuss these aspects in more detail in reference to some of the recent work (doi:10.1021/acssynbio.7b00172). I also encourage the

authors to discuss possible strategies to calibrate the dose response curve of the chimeric TFs, and explain how this would fit into their proposed pipeline. For sensing extracellular compounds, binary outputs could be enough (eg in environmental sensing applications). Other applications, however, such as dynamic control or flux re-direction, need more fine-grained readouts of intracellular metabolite abundance. It would be great to see some further discussions on these aspects as well.

Reviewer #3 (Remarks to the Author):

The manuscript by Juárez et al describes a methodology for assembly of DNA parts combined with custom synthesis of oligo libraries to generate novel chimeric transcription factors that act as repressors of expression in E.coli that are alleviated by addition of benzoate. The manuscript describes an inventive and complicated assembly approach that generates a large library of these chimeric transcription factors and a flow-cytometry based screening method to enrich the library for factors that act as desired - i.e. they repress expression from their cognate promoters in normal E.coli growth conditions but then show accumulation of expression (GFP) when the benzoate inducer is added.

From the title of this manuscript I had expected description and validation of a (somewhat) universal approach to generate small-molecule inducible TF-promoter pairs in E.coli but this is narrowed down to just building and sensing for benzoate and no other inducer. No evidence is presented that this approach could work for anything other than benzoate which means it is hard to say that this is a broadly-applicable strategy at this stage (although it looks promising and generally makes sense). However, this is not my major concern with this work. My major concern is that the quantity of obtained data shown in the paper, and in particular in the main manuscript figures, is miniscule. Figure 1 describes that 135660 combinations are possible through the combinatorial assembly methodology but very limited experimental data is provided for only 2 of these in the end in Figure 4. Even in the supplementary section, there is limited data on anything related to the study. Effectively, what the figures are showing are illustrations of the approach and a description of the methodology.

In that regards I think this work is sadly not yet ready for publication at this stage. I appreciate that the quantity and scope of the work done up until now is impressive, but it feels like what we are seeing here is just the first half of a study. Indeed, most of the paper is a methods description. The authors may want to consider that the extensive efforts to develop eLCR may actually be its own publication at this stage.

So unfortunately I do not recommend publication of this work at this stage unless substantial extra data is shown that verifies the approach and also compares to data obtained at all stages using appropriate controls. This data may already exist so I encourage the authors to consider a resubmission once more data-focused figures can be provided that show evidence of the assembly efficiencies, the flow cytometry sorting effects and how several (not just two) of the chimeric regulators activate gene expression in various conditions compared to known controls. Please also use normal methods to show flow cytometry data (e.g. population histograms). I've never seen a violin plot used for flow data.

REVIEWERS FEEDBACK:

Reviewer #1

We thank Reviewer #1 for the insightful feedback and helpful suggestions to improve the manuscript.

*Although it would be hard to switch terms midway through a project, I think **the authors should reconsider the name of “substrate” binding domain (SBD).** To me, “substrate” has a very specific definition of a ligand undergoes enzymatic catalysis; this process does not occur in these transcription regulators. Alternative names commonly used for these types of proteins include **“ligand” binding domains** (although DNA is also a ligand, so it isn't perfect) and “regulatory” domains (because small molecule binding to this domain regulates DNA binding of the other domain).*

- The manuscript has been modified to substitute *Substrate Binding Domain (SBD)* by *Ligand Binding Domain (LBD)* as suggested.

In several places, paragraphs become very long and could be split (for example on page 11 and in Methods).

- Manuscript has been extensively rewritten to improve readability.

Line 628 contains a possible typo in the phrase “in a subsequent cycle of denaturalization”. Should the last word be “denaturation”?

- This line has been amended (Methods, Line 115).

Reviewer #2

We thank Reviewer #2 for their thoughtful comments and useful suggestions to help us improve the manuscript.

I missed more discussions on the tunability of the biosensors and their expected dose response curve the benzoate sensor was tested for a single concentration only. So far most metabolic engineering applications have focused on metabolite responsive TFs to sense intra or extracellular metabolites. The dose response curve of these can be rapidly tuned with edits to the promoter sequence or the expression level of the TF. Can the same level of tunability be achieved with the chimeric TFs proposed in this paper? The authors should discuss these aspects in more detail in reference to some of the recent work (doi:10.1021/acssynbio.7b00172). I also encourage the authors to discuss possible strategies to calibrate the dose response curve of the chimeric TFs, and explain how this would fit into their proposed pipeline. For sensing extracellular compounds, binary outputs could be enough (eg in environmental sensing applications). Other applications, however, such as dynamic control or flux redirection, need more fine grained readouts of intracellular metabolite abundance. It would be great to see some further discussions on these aspects as well.

- The manuscript has been thoroughly revised and re-written to more clearly state the novelty of the chimeragenesis system and detail how the generation of new transcription factors (TFs) could be a powerful new tool for synthetic biology. Natural transcriptional regulators and newly designed TFs can be integrated into this pipeline, followed by transcriptional and translational fine-tuning or protein engineering of TF binding pockets to focus on delivering the most effective biosensing genetic circuitry.
- In several instances, we have referenced the publication [doi:10.1021/acssynbio.7b00172](https://doi.org/10.1021/acssynbio.7b00172) (reference 18) as an example on how researchers

may improve the properties of TFs through optimization of dose-response curves and protein expression . The manuscript has been amended to include new paragraphs:

Main Text, Lines 50 to 54:

“Our emphasis is on the generation of new TFs capable of detecting small molecules. It is noteworthy, however, that key aspects for the fine tuning of their expression, as well as the refinement of their dose-response curves and ligand affinity, are not tackled in this study. Nevertheless, the products of our assembly and enrichment process are the ideal substrate for systematic expression improvement strategies^{18,19}.”

Main Text, Lines 327 to 339:

“This pipeline represents an enabling first step towards construction of synthetic regulatory circuits. Nevertheless, to achieve their full potential when addressing real-life biotechnological problems, new TFs will greatly benefit from further progress in fine tuning of gene expression. It is well known that the expression of any transgene, and in particular TFs, brings with it a potentially deleterious metabolic burden for the host cell⁴⁷. Fortunately, recent publications have increased our understanding of the delicate underpinnings that permit fine-tuning of gene expression^{18,19,64}. Owing to these advances, we can modulate dynamic range, threshold, and ligand affinity of TFs without first resorting to protein engineering (e.g. modifying promoter strength, operator boxes and RBS). In this context novel strategies, such as the one presented here, can focus more on obtaining TFs capable of recognizing new ligands and less on improving their ligand affinities and dynamic ranges. Dedication to the construction of new biosensors is especially relevant in the Synthetic Biology field, given the limited availability of bio sensing modules¹⁸. Conversely, as new TFs are constructed, there will be an increase in number of protein chassis available to facilitate the engineering of binding pockets for new ligands^{65,66}”

Reviewer #3

We thank reviewer #3 for their candid input and suggested improvements to the manuscript.

From the title of this manuscript I had expected description and validation of a (somewhat) universal approach to generate small molecule inducible TF promoter pairs in E.coli but this is narrowed down to just building and sensing for benzoate and no other inducer. No evidence is presented that this approach could work for anything other than benzoate which means it is hard to say that this is a broadly applicable strategy at this stage [...]

- Reviewer #3 makes a valid point regarding the wider applicability of the system presented in the manuscript. The highest priority of the authors is to present this work as proof-of-principle to the scientific community. Our goal here is to provide a straight-forward and comprehensive description of the chimeragenesis technology, and detailed information on its application (*Methods, Supplementary Materials*), so that it can be reproduced and become an enabling technology for researchers in multiple fields.
- We are confident in the wider applicability of the technology and in collaborative efforts we are now incorporating the technologies into novel biosensors for both environmental and oral-health related applications.
- This work demonstrates the successful construction of functional benzoate-inducible TFs, and therefore validates the technology. In addition, when validating the enrichment system, we obtained emergent glucose-inducible TFs as a by-product of the process which may have improved glucose-sensing ability compared

to SLCP_{GL}. This further supports our hypothesis that PBP can be harnessed for chimeric TFs to effectively recognize soluble small molecules.

- We agree with Reviewer #3 that it will be important to expand our pipeline to create TFs that are responsive to molecules other than benzoate and glucose. Nevertheless, despite the streamlined appearance of the process to construct ChTFBz01 and ChTFBz02, our team has been developing this technology for more than 5 years. We developed the novel cloning method to leverage inexpensive microarray DNA synthesis (eLCR), created an enrichment system based on the construction of a whole family of reporter plasmids, and characterized the complex TF libraries for benzoate and glucose.

Additionally, the need for novel TFs in synthetic biology applications is growing and the authors are aware of multiple groups working to expand the currently available panoply of transcriptional regulators.

We feel that further expansion to other molecules is beyond the scope of this work.

- The manuscript has been modified to more clearly state our objectives and to reflect the potential of this system:
 - a) We increased the emphasis on the ability of the system to enrich glucose-sensing TFs.

Main Text, Lines 226 to 230:

“LacI-GGBP-OD was considerably more abundant in our libraries after enriching for glucose-induced TFs, both when this chimera was added to the pool exogenously and when it was assembled as another fusion-gene present in the library with benzoate-binding chimeras (Supplementary Materials).

These results strongly support the applicability of our chimeragenesis process, described herein, for development of tailor-made TFs able of detecting small molecules of interest.”

Main Text, Lines 316 to 320:

“In this study, we describe a comprehensive strategy to create custom monogenic biosensors using fusion of modular components. Using this strategy, we could independently replicate the construction of a previously characterized chimeric glucose-sensing TF (SLCP_{GL}²⁷) as well as construct multiple novel benzoate-sensing TFs and further characterize two of them: ChTFBz01 and ChTFBz02.”

Supplementary Materials, Lines 590 to 599:

“The inclusion of LacI-GGBP-OD as a built-in positive control included into the assembly of the library was designed so that GGBP was treated as one more of the LBD. All oligonucleotides necessary for the construction of GGBP-based chimeras were included. 285 different “core chimeras” could be glucose-responsive (9044 total different constructions). The overall population of GGBP bearing chimeras in the starting library AYC Lib-Ch-OD should have been close to 6.67% of the total (4522 out of 67830) but was instead 0.09%, increasing to 32.6% after the enrichment process. These data suggest it is possible to find a better glucose-responsive TF (in terms of dynamic range or inducibility) among non-LacI-GGBP-OD chimeras. Glucose biosensors were not the focus of this publication but in the future, we plan to explore a selection of those glucose-sensing TFs in depth. This observation highlights the strength of the chimeragenesis system presented in this work.”

- b) We have edited the conclusion section to clarify the objectives of this work.

Main Text, Lines 320 to 336:

“The novel sensor proteins developed in this work expand the limited collection of available transcriptional repressors that can be used as biosensors in the degradation of lignin. Beyond their immediate usability to tackle this biotechnological problem, they represent an important milestone for the construction of synthetic TFs on demand, as our pipeline can be readily applied to create many other custom-made chimeric TFs. The use of periplasmic binding proteins as detection domains underscores the potential of this method for generating tailored biosensors, and goes far beyond the swapping of domains between currently known regulators.”

[...] My major concern is that the quantity of obtained data shown in the paper, and in particular in the main manuscript figures, is miniscule. Figure 1 describes that 135660 combinations are possible through the combinatorial assembly methodology but very limited experimental data is provided for only 2 of these in the end in Figure 4. Even in the supplementary section, there is limited data on anything related to the study. [...]

- We thank Reviewer #3 for this helpful suggestion. The text has been edited and modified to introduce additional data enabling a more thorough understanding of the work.

Main Text, *Figure 1* has been amended to introduce the percentage of chimeric TFs assigned to the different categories of chimeras for their assembly as explained in Main Text and Supplementary materials.

Supplementary Materials, *Figure S1 (new figure)* represents the true size distribution of the designed chimeric TF genes compared to their abundance in the library. This figure is introduced early in the Main Text (Line 151) helping the reader to visualize the size of the genes encoding the TFs designed in this work, as well as the actual size distribution of the genes recovered after the enrichment process.

Supplementary Materials, *Figure S5 (new figure)* represents the relative abundance of chimeric TFs in the sorted populations after undergoing enrichment for benzoate recognition. This new figure supports the assertion presented in Main Text that after the enrichment process there was a diverse group of new potential benzoate-sensing constructs, rather than a handful of highly abundant chimeric TFs. Main Text (Lines 239-240) *“No single chimera dominated the population, but on average 50 chimeras were present in the enriched libraries in an abundance equal or greater than 0.5%.”*

I encourage the authors to consider a resubmission once more data focused figures can be provided that show evidence of the assembly efficiencies [...]

- We thank Reviewer #3 for this helpful suggestion to improve the manuscript. The text has been edited and modified to introduce relevant information on the efficiency of the construction process.

Main Text, *Figure 2* has been amended to introduce the percentages of chimeras belonging to each assembly class. This information is now displayed within a key figure of the Main Text.

Supplementary Materials, *Table S4 (new table)*. This new table has been added to more clearly show the estimation of the cloning efficiency of libraries and the distribution of cloned TFs among the different classes of chimeras

Main text, Lines 170-177.

A new paragraph has been added to the manuscript to include relevant insights on the outcome of library construction:

“To estimate the composition of the library we sequenced a representative sample of the chimeric TFs within in E. coli (Ch-END) and E. coli (Ch-OD)

libraries (Methods). Table S4 summarizes the assignment of fusion genes contained in both libraries to the different classes of chimeras described in Figure 2 and Supplementary Materials. Chimeric TF genes containing every DBD and LNK were found in different abundance. However, four LBDs did not integrate into any chimeras (ADP71087_nSP, AHF85493_nSP, CAK09396_nSP and KAI94709_nSP). We were unable to discern whether these were disfavored in the cloning process or if the fact that these four LBD lacked their signal peptide increased the toxicity of the chimeric TFs after each had integrated.”

Methods, Lines 101-104.

This paragraph has been amended to include data on the amplification efficiency of the *Infra* and *Supra* oligonucleotides used for the eLCR assembly:

“Interestingly, the amplification yield of Infra was consistently lower than that of Supra (Infra was on average 77.3±13.4% of Supra), especially when the templates were complex libraries; suggesting this was not a result of inferior performance by the primer set, but instead a more intrinsically difficult template library.”

Methods, Lines 120-125.

This paragraph has been amended to include data on the expected yield for the amplification of *Infra* and *Supra* oligonucleotides:

“Given the average proportion between Supra and Infra oligonucleotides and their respective flanking adapters, as well as the fact that dsDNA enters the Type IIS digestion-denaturalization-USER digestion process and ssDNA is retrieved, the maximum possible yield is close to 30% in terms of mass, while we experimentally observed an average 5.1±2.2%. This 6-fold loss can be attributed to the different purification steps. The amplified material was enough for the correct performance of the assembly reactions described below.”

I encourage the authors to consider a resubmission once more data focused figures can be provided that show evidence of [...] the flow cytometry sorting effects [...]

- Reviewer #3 makes a valid point emphasizing the importance of flow cytometry data for the proof-of-concept presented in this manuscript. We agree that the inclusion of this data is of the uttermost importance. We have included extensive information regarding the FACS in our re-written manuscript.

Supplementary Materials, *Figure S5 (new figure)* displays the distribution of the most abundant chimeric TFs obtained after several rounds of flow cytometry enrichment of libraries AYC Lib-Ch-END and AYC Lib-Ch-OD. We observed in both cases that there appears to be no strong sorting effect favoring the capture of a reduced number of chimeric TFs.

Supplementary Materials, *Table S5* indicates the percentages of cells recovered after each enrichment sorting (performed following the parameters detailed in *Methods*).

Supplementary Materials, Lines 517-521 shows the correlation between the expected distribution of a control TF (wtLacI) and the actual data observed after the sorting process.

Supplementary Materials, Lines 548-599 include the percentages of GFP-positive cells that were sorted as well as the percentage of the control chimera SLCP_{GL} (LacI-GGBP-OD) detected after several sorting rounds.

- The Methods section detailing FACS experiments (Lines 304-309) was written in conjunction with Dr. J.K. Moore, Flow Cytometry Director (Department of Systems Biology, Harvard Medical School) to clearly detail all the parameters necessary to replicate the flow cytometry sorting experiments presented in this

work. Dr. Moore also recommended the inclusion of *Figure S6* to illustrate a standard FACS enrichment as performed in this work.

I encourage the authors to consider a resubmission once more data focused figures can be provided that show evidence of [...] how several (not just two) of the chimeric regulators activate gene expression in various conditions compared to known controls.

- We thank Reviewer #3 for their thoroughness. The most studied transcriptional repressors related to benzoate metabolism do not recognize benzoate as their inducer (e.g. BzdR and BoxR, Main Text, Line 301) or it is unknown if benzoate is their inducer (e.g. BamVW, BadM, BgeR, Main Text, Lines 304-305). In the case of BenM (Main Text, Line 312), this TF is a transcriptional activator with a complex inducer landscape. With this information at hand we decided to study the newly constructed chimeras in isolation, due to a lack of close references.
- We reiterate the aforementioned statement on the time and effort already invested in this project. We consider that the work necessary to construct and test more benzoate-sensing chimeric TFs will not add any significant value to the manuscript beyond augmenting the sheer numbers of chimeric TFs validated in this work.
- Based on the enrichment data obtained after several flow cytometry sortings, *Figure S5* suggests that the possibilities to detect a benzoate-sensing TF that represents a radically improvement compared to ChTFBz01 and ChTFBz02 are reduced.

Please also use normal methods to show flow cytometry data (e.g. population histograms). I've never seen a violin plot used for flow data.

- The authors understand that Reviewer #3 interpreted *Figure 4* as a representation of flow cytometry data, however this is not the case. We have included in the figure footnote a statement to clarify this:
“Time course showing relative GFP fluorescence of AYC ChTFBz01 and AYC ChTFBz02 strains grown in minimal medium in a multi-well plate reader as indicated in Methods.”
- To further clarify the origin of data relative to the individual assessment of the functionality of ChTFBz01 and ChTFBz02 we have amended the manuscript:
 Main Text, Lines 267-269: *“In both cases, there was a reduction of the GFP fluorescence of the culture when the chimeras were expressed in regular culture medium (chimeras expressed, GFP promoter repressed), and tracked in a plate-reader as detailed in Methods.”*
 Methods, Line 338 has been modified to remark the use of a plate reader: *“In vivo assay of ChTFBz01 and ChTFBz02 activity (plate reader)”*
- Additionally, *Figure 4* has been used effectively to display the activity of ChTFBz01 and ChTFBz02 in several meetings, including the international conference (17th Annual Mark Wilson Conference - 28th Annual Meeting of the Oral Immunology/Microbiology Research Group; February 2018; Cancun, Mexico).

REVIEWERS' COMMENTS:

Reviewer #2 (Remarks to the Author):

The authors have addressed the referees' comments satisfactorily.

Reviewer #3 (Remarks to the Author):

I was disappointed to realise that I was the minority critic in the first round of peer review for this paper. I hate being that guy! Perhaps I am missing something because the other two reviewers were much more positive about this work. I will therefore try to find a way to be more accepting of it now.

Mostly I still find it odd that the authors are keen to publish an experimental paper with 4 figures where only the final figure shows any experimental data at all. I know that there is some supplementary data and more has now been added, but the fact that none of this makes the main figures is strange. It is actually illustrative of the fact that this is more of a 'concept' study that outlines a method that could potentially be used to make many different metabolite biosensors, and is verified as a workable concept by in the end making two new biosensors for the one metabolite.

It is clear from the rebuttal that the authors do not want to expend any further time and energy to prove that their approach is good for more than just creating sensors for benzoate (and the accidental glucose case). They are clear in stating that this is a lot of work despite their new method and all the libraries being made by them already. This is disappointing of course, but if the paper does not sell itself as a generalisable method, then it is true that there is no need to prove it with such further work.

Therefore I would be okay to see this published if the authors make a proper attempt to tweak the text in the title, abstract and introduction to make it clearer that this work only presents a 'concept' for generating biosensor transcription factors that is validated here only for the case of benzoate detection. I notice in the rebuttal to my review that the authors avoided changing any text in the first half of their paper to deal with my original point on this. The reason I was so let down first time around was because the title, abstract and intro had built-up the work to be an awesome new general method, but then I got to the end and was only ever shown a sensor for benzoate. Appropriate changes at the start of the paper can hopefully prevent the work being over-promised and stop future readers also being let down as they get to the end of the manuscript. I suggest changing the title.

On the other point, if the authors (and editor) feel that it is fine to present only 1 experimental result in all their main figures then I guess that's their choice.

Reviewer #3

We thank reviewer #3 for the thoughtful comments and useful suggestions to improve this manuscript.

Therefore I would be okay to see this published if the authors make a proper attempt to tweak the text in the title, abstract and introduction to make it clearer that this work only presents a 'concept' for generating biosensor transcription factors that is validated here only for the case of benzoate detection. [...] Appropriate changes at the start of the paper can hopefully prevent the work being over-promised and stop future readers also being let down as they get to the end of the manuscript.

- Reviewer #3 makes a valid point regarding the broad applicability of the system presented in the manuscript. We have modified the first half of the manuscript to clarify the scope for the readers, and included that the method has been currently validated with two benzoate-sensing transcription factors.
- To emphasize the experimental nature of this work for the generation of novel biosensors, we have substituted the word “*strategy*” with “*concept*” as early on as in the Abstract.

Main Text, Abstract, Line 24:

“We validate this concept by constructing and functionally testing two unique sense-and-respond regulators for benzoate.”

An additional two new sentences were added to the Introduction to include that the pipeline has been validated exclusively with two TFs for a chemical compound.

Main Text, Introduction, Lines 56-57:

“This approach is validated by the generation of two new benzoate-binding TFs.”

Main Text, Introduction, Lines 76:

“The two novel TFs presented here...”

I suggest changing the title.

- We thank Reviewer #3 for the thoroughness in the analysis of the paper. The authors have collectively discussed at length a possible change in the title. In our opinion “*Biosensor libraries harness large classes of binding domains for construction of allosteric transcriptional regulators*” is a concise and descriptive assertion for this proof of concept work. Unfortunately, Reviewer #3 has not suggested a specific alternative and we have therefore kept this title in place.